# A preliminary indication that HLA-A*03:01 may be associated with visceral leishmaniasis development in people living with HIV in Ethiopia

Nicky de Vrij[1,2], Romi Vandoren[2], Kadrie Ramadan[3], Anke Van Hul[1], Ann Ceulemans[4], Mekibib Kassa[5], Roma Melkamu[5], Arega Yeshanew[5], Tadfe Bogale[5], Hailemariam Beyene[6], Kasaye Sisay[6], Aderajew Kibret[6], Dagnew Mersha[6], Wim L. Cuypers[2], Florian Vogt[7,8,9], Saskia van Henten[7], Koert Ritmeijer[10], Thao-Thy Pham[1], Pieter Meysman[2], Kris Laukens[2], Bart Cuypers[2], Ermias Diro[5,11], Rezika Mohammed[5], Johan van Griensven[7], Wim Adriaensen[1] *

1 Clinical Immunology Unit, Department of Clinical Sciences, Institute of Tropical Medicine, Antwerp, Belgium, 2 Adrem Data Lab, Department of Computer Science, University of Antwerp, Antwerp, Belgium, 3 Clinical Reference Laboratory, Department of Clinical Sciences, Institute of Tropical Medicine, Antwerp, Belgium, 4 Virus Ecology, Department of Biomedical Sciences, Institute of Tropical Medicine, Antwerp, Belgium, 5 Leishmaniasis Research and Treatment Centre, University of Gondar, Gondar, Ethiopia, 6 Médecins Sans Frontières, Abdurafi, Ethiopia, 7 Unit of Neglected Tropical Diseases, Department of Clinical Sciences, Institute of Tropical Medicine, Antwerp, Belgium, 8 National Centre for Epidemiology and Population Health, The Australian National University, Canberra, Australia, 9 The Kirby Institute, University of New South Wales, Sydney, New South Wales, Australia, 10 Médecins Sans Frontières, Amsterdam, The Netherlands, 11 Department of General Internal Medicine, University of Washington, Washington, United States of America

* wadriaensen@itg.be

**Data Availability Statement:** The data and scripts used to perform the HLA association analyses and create the figures in this manuscript have been

## Abstract

Human immunodeficiency virus (HIV) co-infection is a major challenge for visceral leishmaniasis (VL) control, particularly in Ethiopia where the incidence of both pathogens is high. VL-HIV often leads to high rates of antileishmanial treatment failure and recurrent VL disease relapses. Considering the high prevalence of HIV and *Leishmania* in the Ethiopian population, preventing the progression of asymptomatic *Leishmania* infection to disease would be a valuable asset to VL disease control and to the clinical management of people living with HIV (PLWH). However, such a strategy requires good understanding of risk factors for VL development. In immunocompetent individuals living in Brazil, India, or Iran, the Human Leukocyte Antigen (HLA) gene region has been associated with VL development. We used NanoTYPE, an Oxford Nanopore Technologies sequencing-based HLA genotyping method, to detect associations between HLA genotype and VL development by comparing 78 PLWH with VL history and 46 PLWH that controlled a *Leishmania* infection, all living in a VL endemic region of North-West Ethiopia. We identified an association between HLA-A*03:01 and increased risk of VL development (OR = 3.89). These data provide candidate HLA alleles that can be further explored for inclusion in a potential *Leishmania* screen-and-treat strategy in VL endemic regions.

deposited on Zenodo (https://doi.org/10.5281/zenodo.12516076). Raw sequencing data cannot be shared publicly because of institutional restriction to publishing information that could be used to de-identify individuals in the study. Data are available from the ITM institutional data access committee for researchers who meet the criteria for access to confidential data. This committee can be contacted at the following email address: ITMresearchdataaccess@itg.be.

**Funding:** This work was supported by the Research Foundation Flanders (FWO) [1S71721N fellowship to N.d.V.], a research grant of the University of Antwerp Research Fund (BOF) [FFB220027 to B.C. and N.d.V.], the Institute for Tropical Medicine Antwerp's SOFI programme supported by the Department of Economy, Science and Innovation of the Flemish Government, and the Belgian Directorate General for Development Cooperation under the ITM-DGDC framework agreement FA-III & FAIV. The funders had no role in study design, data collection and analysis, decision to publish, or preparation of the manuscript. The funders had no role in study design, data collection and analysis, decision to publish, or preparation of the manuscript.

**Competing interests:** I have read the journal's policy and the authors of this manuscript have the following competing interests: W.A. received a travel grant by Omixon to participate in the European Federation for Immunogenetics conference to present the preliminary results of this work. Omixon had no role in study design, data collection and analysis, preparation of the manuscript, nor the decision to publish.

## Author summary

Human immunodeficiency virus (HIV) co-infection is a major challenge for the control of visceral leishmaniasis (VL), particularly in Ethiopia where both frequently occur. VL-HIV disease is often hard to treat, and some treated patients will relapse after initially thought to be cured. However, there is a long period of asymptomatic *Leishmania* infection prior to the development of VL-HIV symptoms, and tackling the infection at this stage will be valuable in the effort to combat VL-HIV disease. However, this requires good understanding of human risk factors for VL development. In this work, we tested the Human Leukocyte Antigen gene region and the association with VL disease development, as genetic risk factors. We identified a preliminary association between the HLA-A*03:01 gene variant and an increased risk for VL development. This would need to be tested again in a larger study, to assess whether it can be used to see if asymptomatic *Leishmania*-infected individuals will develop VL disease.

## Introduction

Visceral leishmaniasis (VL) is a potentially fatal disease caused by protozoan parasites of the *Leishmania donovani* complex and is transmitted by the bite of an infected sand fly. Although the global annual incidence of VL has dropped from an estimated 50.000–90.000 cases in 2016 to around 13.000 cases in 2020, Ethiopia continues to be highly burdened and, together with other countries of the East African region, hosts up to 60% of all global VL cases [1]. In North-West Ethiopia, between 25% and 40% of VL patients are co-infected with HIV (VL-HIV) which presents a substantial challenge to VL control [2,3]. For instance, VL treatment outcomes are often worse in VL-HIV patients compared to immunocompetent VL patients. While up to 90–95% of immunocompetent VL patients cure after treatment, HIV co-infected individuals frequently fail treatment with up to 30% treatment failure rates [2]. Moreover, up to 60–80% of VL-HIV patients will develop recurrent disease episodes compared to around 5% of immunocompetent VL patients [2,4].

Considering the high prevalence of HIV and *Leishmania* infection in the Ethiopian population, preventing the progression of asymptomatic *Leishmania* infection to disease would be a valuable asset to VL disease control and to the clinical management of people living with HIV (PLWH) [5,6]. The asymptomatic stage preceding VL disease is detectable by a variety of *Leishmania* infection markers, and can thus provide an opportune moment to screen for those at risk for VL development and to initiate preventative strategies [5–7]. Such a preventative strategy is already successfully applied in other infectious disease settings, For example, primary preventative therapy is recommended for other opportunistic infections in HIV such as in cryptococcal disease or tuberculosis if individuals are positive for markers of early infection [8–10]. However, only the minority of *Leishmania*-infected individuals progress to disease, and it is currently not fully known what predisposes these individuals to progress to VL disease in the context of a HIV co-infection. A screen-and-treat approach would require a clinical algorithm that incorporates biomarkers with predictive value to detect those individuals at high risk for VL development [5]. While predictive biomarkers for primary VL development in a HIV co-infection setting have not been identified, several predictive factors have been identified for VL relapse in PLWH. These include a low CD4+ cell count ($<200/mm^3$), the inability to reconstitute these CD4+ cell counts upon antiretroviral therapy (ART) initiation, not being on ART at time of VL development, a history of prior VL episodes, initial VL

treatment failure, high parasite loads at the time of VL diagnosis, and *Leishmania* antigenuria [11–13]. However, whether these factors also influence susceptibility for primary VL development in HIV co-infected individuals is not fully understood. Yet, overlap between VL development and VL relapse factors is expected to be considerable, as a low CD4+ cell count and high *Leishmania* antigenuria at time of primary VL diagnosis can often be observed [13,14].

Host genetic susceptibility factors are of particular interest in the context of a clinical algorithm to capture those at risk, as they remain constant over a person's lifetime and would only need to be determined once. The strongest reported genetic susceptibility factor for VL development is the association between the Human Leukocyte Antigen (HLA) gene region and VL development, which can affect the disease susceptibility both positively and negatively [15–18]. The HLA gene region is located on the short arm of chromosome 6 at band 21 and is the most polymorphic human genomic region, encoding for distinct Major Histocompatibility Complex (MHC) molecules [19]. MHC molecules present pathogen-derived antigens to cognate T cells to elicit an immune response. An individual can carry a maximum of 12 classical HLA gene variants, or alleles, on a chromosome. Currently, three studies have shown strong associations between VL development in immunocompetent individuals and a total of 9 HLA alleles, with odd's ratios ranging from 0.42 to 1.76, suggesting possible prognostic value [15–18]. It is important to note that these studies have been carried out in India, Brazil, and Iran, and no such study has been performed for the highly VL-endemic East African region [15]. However, as HLA genotype varies substantially across geographically and ethnically, and *Leishmania* parasite genetics also differ across continents, results from studies in other regions can not simply be extrapolated. Therefore, identifying HLA associations in the highly VL-endemic East-African region as well is of particular importance.

Thus, in this work, we employed Nanotype, a rapid and mobile Oxford Nanopore Technologies (ONT) sequencing-based HLA genotyping method, to detect associations between HLA genotype and VL development by comparing 78 HIV patients with VL history and 46 HIV patients that controlled a *Leishmania* infection, all living in a VL endemic region of North-West Ethiopia.

## Material and methods

### Ethics statement

This study was approved by the Ethiopian National Research Ethics Review Committee (16/24/253), the University of Gondar Institutional Review Board (V/P/RCS/05/708/2017), and the Institute of Tropical Medicine Antwerp Institutional Review Board (1091/16). Written consent was obtained from patients for long-term biobanking and future research on their biobanked samples, including testing of genetic material.

### Study population and design

This case-control study was performed on biobanked samples of the PreLeisH cohort study conducted in the Abdurafi region of Amhara in North-West Ethiopia (clinicaltrials.gov NCT03013673), in which 570 adult HIV-positive participants were enrolled between October 2017 and May 2021, and were followed up at three to six months intervals to monitor *Leishmania* infection and VL development at the MSF-supported Abdurafi Health Centre (AHC). The AHC has an active HIV treatment program in place with stable HIV follow-up care according to national guidelines, which also includes the initiation of ART. The inclusion criteria for the main PreLeisH study were a confirmation of HIV-positivity, and being enrolled in this active HIV treatment program. Exclusion criteria were being under 18 years old, having an active VL diagnosis at time of enrolment, inability to provide informed consent, and a

medical emergency or other chronic medical conditions that made adherence to or participation in the study unlikely or inadvisable. All PreLeisH-enrolled participants that tested positive for a rK39 RDT, and every fifth participant enrolled, underwent blood sampling with 10mL Sodium-Heparin CPT Mononuclear Cell Preparation tubes (BD Biosciences, U.S.A.). Next, PBMCs were isolated from blood and cryopreserved in a freezing solution (40% FBS, 60% RMPI, 10% DMSO) within 4–6 hours after collection. These PBMCs were then shipped to and biobanked at the Institute of Tropical Medicine in Antwerp.

In order to detect associations between HLA alleles and VL development, we included 78 PLWH who had a history of VL prior to the study (past VL group) as cases. Participants with incident VL development (n = 38) during the PreLeisH trial were not included in this ancillary study because their samples were used for primary analyses. To ensure a strong control group that has been exposed to the parasite but resisted VL development, a stringent criterium was chosen due to the lack of a golden standard test for asymptomatic *Leishmania* infection among the PLWH [7]. Thus, participants with positivity for only one *Leishmania* infection marker were excluded (n = 126), and as robust controls (Asymptomatic *Leishmania* controllers group) we included 46 PLWH without VL history who tested positive for at least two *Leishmania* infection markers (rK39 RDT, rK39 ELISA, DAT, a *Leishmania* antigenuria-based latex agglutination test (KAtex), or a real-time PCR targeting presence of *Leishmania* kinetoplast DNA in whole blood) at any time during the study, and who remained asymptomatic during the complete study period (**Fig 1**).

## DNA extraction and quantification

DNA was extracted from the PBMC samples of all participants using the Maxwell RSC 48 instrument (Promega, Wisconsin, U.S.A) using the Maxwell RSC Whole Blood DNA kit (Promega, Wisconsin, U.S.A.) according to the manufacturer's instructions. Next, the concentration of the DNA was measured using the Qubit 1X dsDNA BR assay (ThermoFisher, Waltham, U.S.A.) on a Qubit Fluorometer 4.0 device (ThermoFisher, Waltham, U.S.A.).

## HLA genotyping for HLA association analysis using NanoTYPE

Participant DNA samples were used for HLA genotyping using the ONT sequencing-based NanoTYPE kit (Omixon, Budapest, Hungary). A total of 200ng of DNA was used to prepare the library according to the manufacturer's instructions. In brief, an enrichment PCR was performed using the HLA Multi Primer Mix and reagents included in the NanoTYPE kit. Next, amplicons were quantified using the Qubit 1X dsDNA BR assay (ThermoFisher, Waltham, U. S.A.) on a Qubit Fluorometer 4.0 device (ThermoFisher, Waltham, U.S.A.), and a total of 200ng of amplicon was transferred to a new tube. A barcoding step was then performed using the Rapid Barcoding Plate provided in the Rapid Barcoding 96 Kit (SQK-RBK110.96; ONT, Oxford, U.K.). After this, samples were pooled (between 8–12 samples per library), adding 8.5 µl of barcoded amplicons per sample to a library. Next, resulting pooled libraries were subjected to size selection and purification using the AMPure XP beads (Beckman Coulter, Brea, U.S.A.) also provided in the Rapid Barcoding 96 Kit. A total of 10 µl of purified library was then transferred to a new tube, and 1 µl of Rapid Adapter F (provided in the Rapid Barcoding 96 Kit) was added to the library. The resulting library with added adapters was then mixed with 37.5 µl of Sequencing Buffer II and 25.5 µl of Loading Beads II to load on to a R9.4.1 Flow Cell (ONT, Oxford, U.K.) for subsequent sequencing on a MinION Mk1B or Mk1C instrument (ONT, Oxford, U.K.) with MinKNOW version 22.05.5. Libraries were sequenced for at least 1 hour per sample in a library. The resulting FAST5 files were basecalled using Guppy version 6.1.5 using the high-accuracy model. Next, basecalled FASTQ files were used for HLA genotyping with the NanoTYPER v1.2.0 software (Omixon, Budapest, Hungary).

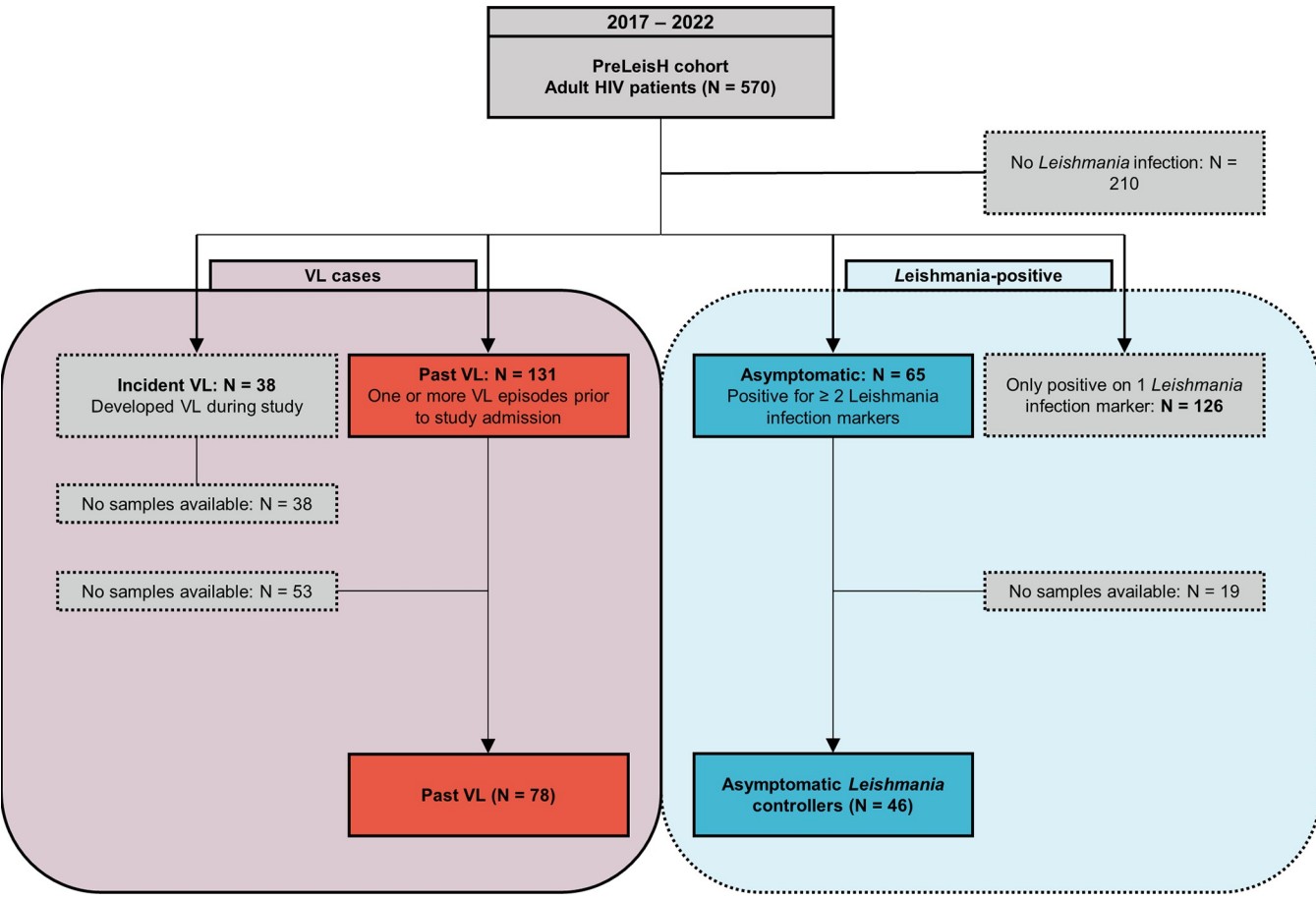

**Fig 1. Flow chart of study participants and inclusion criteria for this ancillary study on biobanked samples of the PreLeisH cohort study.** This ancillary study was performed on biobanked samples of adult HIV-infected participants that were enrolled in the PreLeisH (clinicaltrials.gov NCT03013673) cohort study that ran between 2017 and 2022, for which the inclusion criteria were a confirmation of HIV-positivity, and being enrolled in this active HIV treatment program. The exclusion criteria were being under 18 years old, having an active VL diagnosis at time of enrolment, inability to provide informed consent, and a medical emergency or other chronic medical conditions that made adherence to or participation in the study unlikely or inadvisable. The participants were stably under HIV care at the MSF-supported Abdurafi Health Centre, and were followed up at three to six months intervals to monitor Leishmania infection and VL development. Leishmania infections markers monitored include rK39 RDT, rK39 ELISA, DAT, KAtex, and Leishmania kinetoplast DNA real-time PCR on blood. For this ancillary study, the participants with a prior history of VL (Past VL, in red) were included as cases, and the participants without a prior history of VL but testing positive for at least two Leishmania infection markers at any timepoint during the study were included as controls (Asymptomatic Leishmania controllers, in blue).

## Reference HLA genotyping using the AlloSeq Tx17 kit

In order to assess the ONT-based NanoTYPE kit's accuracy for HLA association analyses and feasibility for low-and-middle income countries, we compared it to the widely used Illumina-based AlloSeq Tx17 kit (CareDx, Brisbane, U.S.A.) for a subset of 24 participants. A total of 500ng of DNA per participant was used to prepare the library according to the manufacturer's instructions. In brief, a tagmentation step was performed followed by an indexing PCR step to barcode the samples. Next, samples were subjected to size selection and purification using purification beads and pooled into two equal-sized libraries. The two library pools were implemented in a hybridization PCR step with the AlloSeq Tx17 probe panel. Following this, the library was purified using a series of capture wash steps. Subsequently, a post-enrichment PCR and clean-up was performed. Finally, the libraries were sequenced on the MiniSeq instrument (Illumina, San Diego, U.S.A.) with the MiniSeq Mid Output 300-cycles Kit (Illumina, San

Diego, U.S.A.) and libraries diluted to 20pM and denatured with a 1% (5pM) PhiX spike-in. Out of the 24 samples, 8 were resequenced by the CareDx company due to low coverage and sequencing dropouts. For these resequenced samples, only results produced by CareDx were included. Resulting FASTQ files were used for analysis with the AlloSeq Assign v1.0.3.1337 software (CareDx, Brisbane, U.S.A.).

### Statistics and HLA association analysis

For participant characteristics, continuous variables were represented as medians with inter-quartile ranges (IQR) and categorical data as numbers and proportions (%). To compare continuous variables across the participant groups, a Mann-Whitney U test was performed after testing for normality with the Shapiro-Wilk test. Statistical testing for categorical variables was performed with a Fisher's exact test. All tests were two-tailed. All statistical analyses were performed using the R Statistical Software at version 4.1.1. An upset plot to display participant *Leishmania* infection marker positivity was made using the UpsetR package at version 1.4.0 for the R Statistical Software at version 4.1.1 [20,21].

To investigate the association between HLA alleles and the development of VL, a Fisher's exact test was applied in Python version 3.12.0 to compare past VL developers to those who are able to control their *Leishmania* infection [22]. Enrichment or depletion of HLA alleles between groups was tested at the 4-digit/2nd field allele resolution (e.g. *HLA-A*01:01*) and for 8 distinct HLA genes spanning both HLA class I and class II alleles separately (*HLA-A, -B, -C, -DQA1, -DQB1, -DPA1, -DPB1, and -DRB1*). The effect size of the association between HLA alleles and VL disease was determined using the odds ratio (OR) and 95% confidence interval (95% CI), calculated after adding a pseudocount of 1 to each allele. Alleles only present in less than 5% of the participants ($n < 6$) were not included in the analysis. Benjamini-Hochberg multiple testing correction was applied to control for the false discovery rate (FDR < 0.05). A Hardy-Weinberg Equilibrium test was performed using MiDAS version 1.2.0 in R version 4.1.1 [21,23].

### Reference HLA allele frequency distribution in population

The Allele Frequency Net Database (AFND) was used to compare prior known HLA allele frequencies in the studied population to our findings, in order to get a better understanding of the representability of our cohort and a more holistic view of the HLA background of the studied population [24]. This data was used to assess whether any of the previously described HLA alleles assocations with VL development (reviewed in de Vrij et al, 2021 [15]) occured in our studied population. Only one study was found in which the authors reported on targeted *HLA-DRB1, -DQA1, and -DQB1* allele frequencies in the Ethiopian Amhara population [25]. Allele frequencies reported in our study were calculated using the MiDAS package at version 1.2.0 for the R Statistical Software at version 4.1.1 [23].

## Results

### Participant characteristics and *Leishmania* infection marker positivity

Our study included a total of 124 *Leishmania*-infected participants living with HIV, including 78 (62.9%) participants with past VL and 46 (37.1%) asymptomatic *Leishmania* controllers. The latter group was followed up for a median of 22 (16–23 IQR) months. Participants with a history of VL had a median of 1 (1–2 IQR) VL episode prior to the study, with their most recent VL episode being a median of 79.5 (32–145 IQR) months ago. Out of all past VL developers, 44 (56.4%) were already diagnosed with HIV at time of VL development. Of these 44

**Table 1. Participant socio-demographic characteristics and diagnostic marker positivity at study enrolment.**

|  | Past VL (n = 78) | Asymptomatic *Leishmania* controllers (n = 46) | *P*-value |
|---|---|---|---|
| **Socio-demographic characteristics** | | | |
| Age in years, median (IQR) | 38 (35–45) | 40.5 (32–52) | 0.464 |
| Male, n (%) | 76 (97.4) | 44 (95.7) | 0.627 |
| BMI, median (IQR) | 18.2 (16.9–19.6) | 18.9 (17.5–20.1) | 0.138 |
| Occupation, n (%) | | | 0.378 |
| Daily labourer | 43 (55.1) | 21 (45.6) | |
| Farmer | 24 (30.8) | 13 (28.3) | |
| Other | 10 (12.8) | 11 (23.9) | |
| None | 1 (1.3) | 1 (2.2) | |
| **_Leishmania_ infection marker positivity** | | | |
| rK39 RDT positivity, n (%) | 73 (93.6) | 38 (82.6) | 0.070 |
| rK39 ELISA positivity, n (%) | 63 (80.8) | 29 (63) | 0.035 |
| DAT positivity, n (%) | 64 (82.1) | 22 (47.8) | <0.001 |
| KAtex positivity, n (%) | 2 (2.6) | 3 (6.5) | 0.359 |
| *Leishmania* PCR positivity, n (%) | 5 (6.4) | 3 (6.5) | 1 |

NA = Not Applicable. BMI: body mass index (kg/m2); DAT: RDT: ELISA: ART

VL-HIV patients, 34 (77.3%) were on ART prior to their VL development. All but one (45; 97.8%) of the asymptomatic *Leishmania* controllers were already on ART at study enrolment, and the remaining participant was initiated on ART at time of enrolment, according to national HIV treatment guidelines.

The participants were relatively young (median age of 38 years old) and mostly male (96.8%), with the majority working as daily labourers (51.6%) and farmers (29.8%), occupations shown to have high risk of *Leishmania* transmission (**Table 1**) [26]. All participants were permanent residents of the lowlands region of Abdurafi. At study enrolment, the majority of participants with past VL were positive on a rK39 RDT (93.6%), rK39 ELISA (80.8%), and a DAT (82.1%), while KAtex (2.6%) or PCR (6.4%) positivity was less often observed. Most asymptomatic *Leishmania* controllers were already positive on the rK39 RDT (82.6%) at study enrolment, but in contrast to the participants with past VL, the asymptomatic *Leishmania* controllers were less often positive on the rK39 ELISA (63%) or DAT (47.8%). As seen in **Fig 2A**, during the study, the majority of the asymptomatic *Leishmania* controllers developed a positive signal on the rK39 ELISA (78.3%) and the DAT (80.4%), and a minority became positive on the KAtex (17.4%) and the PCR (8.7%). The majority (64.1%) of the past VL group were co-positive on the rK39 RDT, rK39 ELISA and DAT at any timepoint during the study (**Fig 2B**). The asymptomatic *Leishmania* controller group showed a similar trend, as co-positivity on the rK39 RDT, rK39 ELISA and DAT was most often observed (41.3%) in this group (**Fig 2A**).

## ONT-based HLA genotyping showed high concordance with Illumina-based HLA genotyping

Next, we assessed the accuracy of ONT-based HLA genotyping by measuring the concordance between this assay (NanoTYPE) and a widely used Illumina-based assay (AlloSeq Tx17, CareDx) on a subset (n = 24) of participant samples.

Overall, the concordance between the tested methods was between 97.9% and 100% for all of the tested HLA loci at all field resolutions, except for HLA-DPB1 which had a 83%

**a**

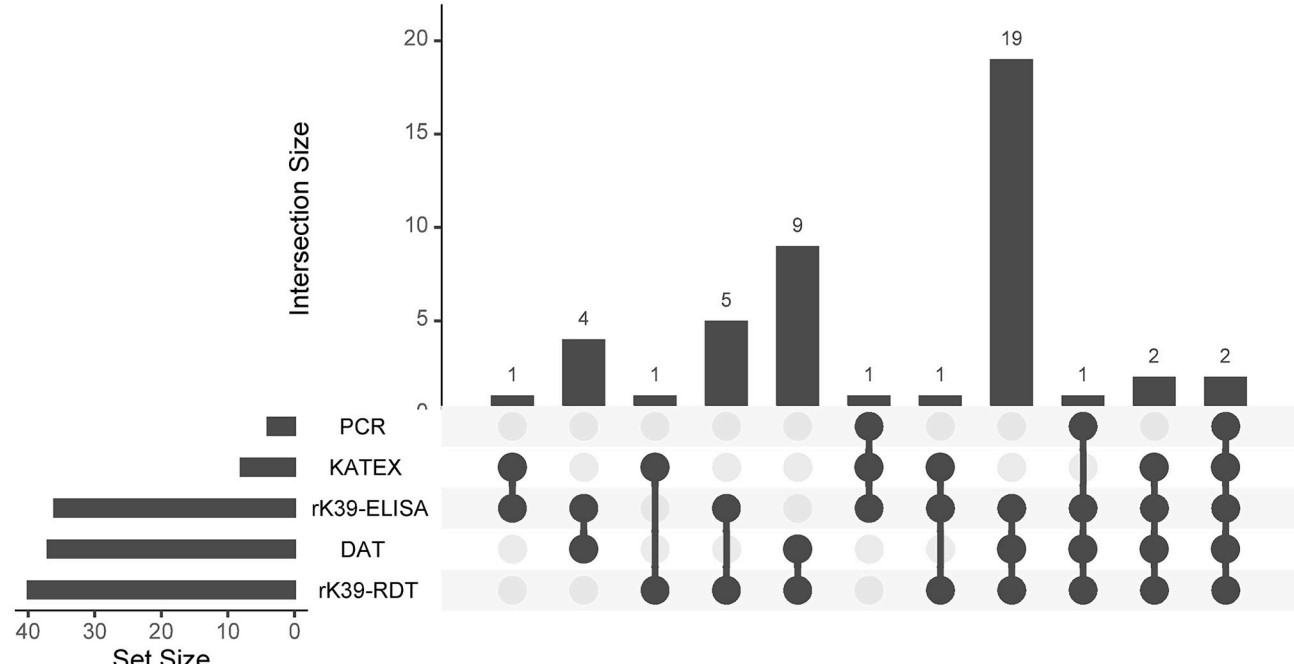

**b**

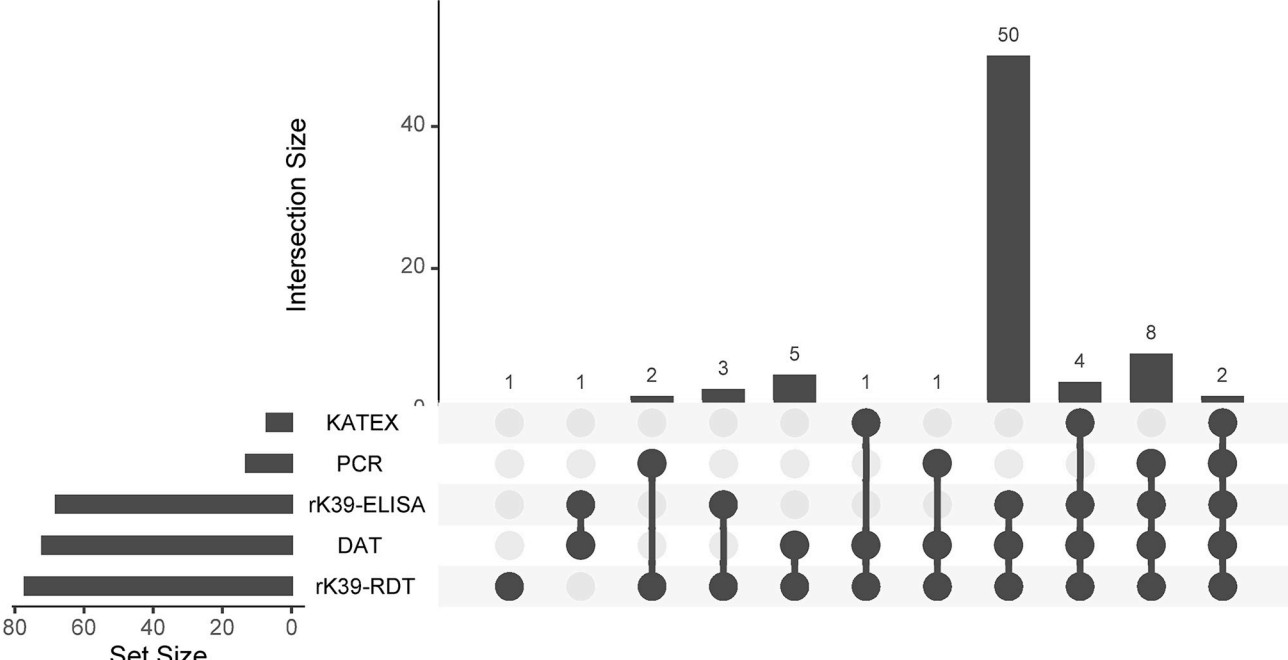

**Fig 2. Upset plot showing positivity on the *Leishmania* infection markers (rK39 RDT, rK39 ELISA, DAT, KAtex, and *Leishmania* PCR) for A) the asymptomatic *Leishmania* controllers (n = 46) and B) the past VL group (n = 78).** Positivity in this graph means positivity at any study timepoint. The set size is the number of participants positive for any of the *Leishmania* infection markers, and the intersection size is the number of participants co-positive for a particular combination of *Leishmania* infection markers.

**Table 2. The concordance between the ONT-based NanoTYPE kit and the Illumina-based AlloSeq Tx17 kit.** Concordance is compared at the first field (2-digit, i.e. HLA-A*01), second field (4-digit, i.e. HLA-A*01:01), and third field (6-digit, i.e. HLA-A*01:01:01), for HLA-A, -B, -C, -DPA1, -DPB1, -DQA1, -DQB1, and -DRB1. N = 24.

| HLA locus | 1st Field Concordance | 2nd Field Concordance |
|---|---|---|
| HLA-A | 100% | 100% |
| HLA-B | 100% | 100% |
| HLA-C | 100% | 100% |
| HLA-DPA1 | 100% | 100% |
| HLA-DPB1 | 83% | 83% |
| HLA-DQA1 | 100% | 100% |
| HLA-DQB1 | 97.9% | 97.9% |
| HLA-DRB1 | 100% | 97.9% |

concordance at all fields (Table 2). This high concordance on the majority of HLA genes indicates sufficient power of the ONT-based NanoTYPE kit for HLA association analyses.

## HLA-A*03:01 is associated with a higher risk of VL development

To assess whether any HLA alleles were associated with past VL development, we performed a HLA association analysis between asymptomatic *Leishmania* controllers and past VL developers. More specifically, we compared alleles across eight classical HLA genes (*HLA-A*, *-B*, *-C*, *-DQA1*, *-DQB1*, *-DPA1*, *-DPB1*, *and -DRB1*) between these participant groups.

We detected a total of 165 unique HLA alleles (up to 4-digit resolution) spanning the eight classical HLA genes in the study population, of which 83 alleles were included in the HLA association analysis because they were present in more than 5% of the participants (n > 6; S1 Table). The HLA allele frequencies of all 165 unique alleles per participant group is shown in S2 and S3 Figs. We observed that HLA-A*03:01 was significantly more frequent in past VL developers (OR = 3.89, 95% CI = 1.62–9.36, $p_{BH}$ = 0.0170, Table 3). In addition to the association between the risk of VL development and HLA-A*03:01, we observed several weaker associations that were significant only before multiple testing correction, including: HLA-DQB1*05:01, HLA-A*02:01, HLA-C*16:04, and HLA-DQA1*01:05 (Table 3). Of these, the HLA-DQB1*05:01 allele was more present in the past VL group (26.9%) than the asymptomatic *Leishmania* controllers (10.9%). Conversely, HLA-A*02:01 was rather linked with a lower risk for VL development as it was present at a lower frequency in past VL developers (16.7%) compared to the asymptomatic *Leishmania* controllers (32.6%). Two alleles, HLA-C*16:04 and HLA-DQA1*01:05, were exclusively present in the past VL developers (9% and 11.5%). Taken together, the association between HLA-A*03:01 and VL development, and the high prevalence of HLA-A*03:01 in the past VL group, suggests prognostic value in capturing those high at risk for VL in a clinical algorithm. In addition to HLA-A*03:01, the weaker associations, in particular those present only in the past VL group such as HLA-C*16:04 and HLA-DQA1*01:05, could have supplementary prognostic value and require further validation.

## Comparison of HLA allele frequencies with population frequencies from prior literature and comparison of detected HLA associations with findings in other continents

In order to get a better understanding of the representability of our cohort, we first performed a Hardy-Weinberg Equilibrium test and next calculated the allele frequencies of all HLA alleles observed in our study population and, in particular for HLA-DQA1, -DQB1, and DRB1, compared them to a reference dataset [15,25]. This reference dataset consists of a targeted HLA

**Table 3. The top 5 HLA alleles detected in the HLA association analysis between past VL developers (n = 78) and asymptomatic Leishmania controllers (n = 46), ranked ascendingly by lowest p-value after Benjamini-Hochberg multiple testing correction.** An OR above 1 indicates an increased risk for VL development, while an OR below 1 indicates increased protection against VL development.

| HLA allele | Past VL (n, %) | Asymptomatic Leishmania controllers (n, %) | OR (95% CI) | p-value | $p_{BH}$-values |
|---|---|---|---|---|---|
| HLA-A*03:01 | 34 (43.6%) | 7 (15.2%) | 3.89 (1.62–9.36) | 0.001 | 0.017 |
| HLA-DQA1*01:05 | 9 (11.5%) | 0 (0.0%) | 6.71 (0.83–54.21) | 0.026 | 0.206 |
| HLA-A*02:01 | 13 (16.7%) | 15 (32.6%) | 0.42 (0.18–0.98) | 0.048 | 0.286 |
| HLA-DQB1*05:01 | 21 (26.9%) | 5 (10.9%) | 2.66 (0.99–7.12) | 0.041 | 0.405 |
| HLA-C*16:04 | 7 (9.0%) | 0 (0.0%) | 5.22 (0.63–43.12) | 0.046 | 0.455 |

OR = Odds Ratio; CI = Confidence Interval; pBH = Benjamini-Hochberg corrected p-values

genotyping of select HLA-DQA1, -DQB1, and -DRB1 genes of the general Amharic population (n = 98) conducted in 1998, here described as the AFND_Amhara cohort [25].

In our analysis, we identified a total of 172 unique HLA alleles (up to 4-digit resolution; 165 unique alleles excluding DRB3/4/5) among 124 individuals. The allele frequency of each allele is shown in **S1 Fig** and provided in **S2 Table**. Our analysis showed no obvious population

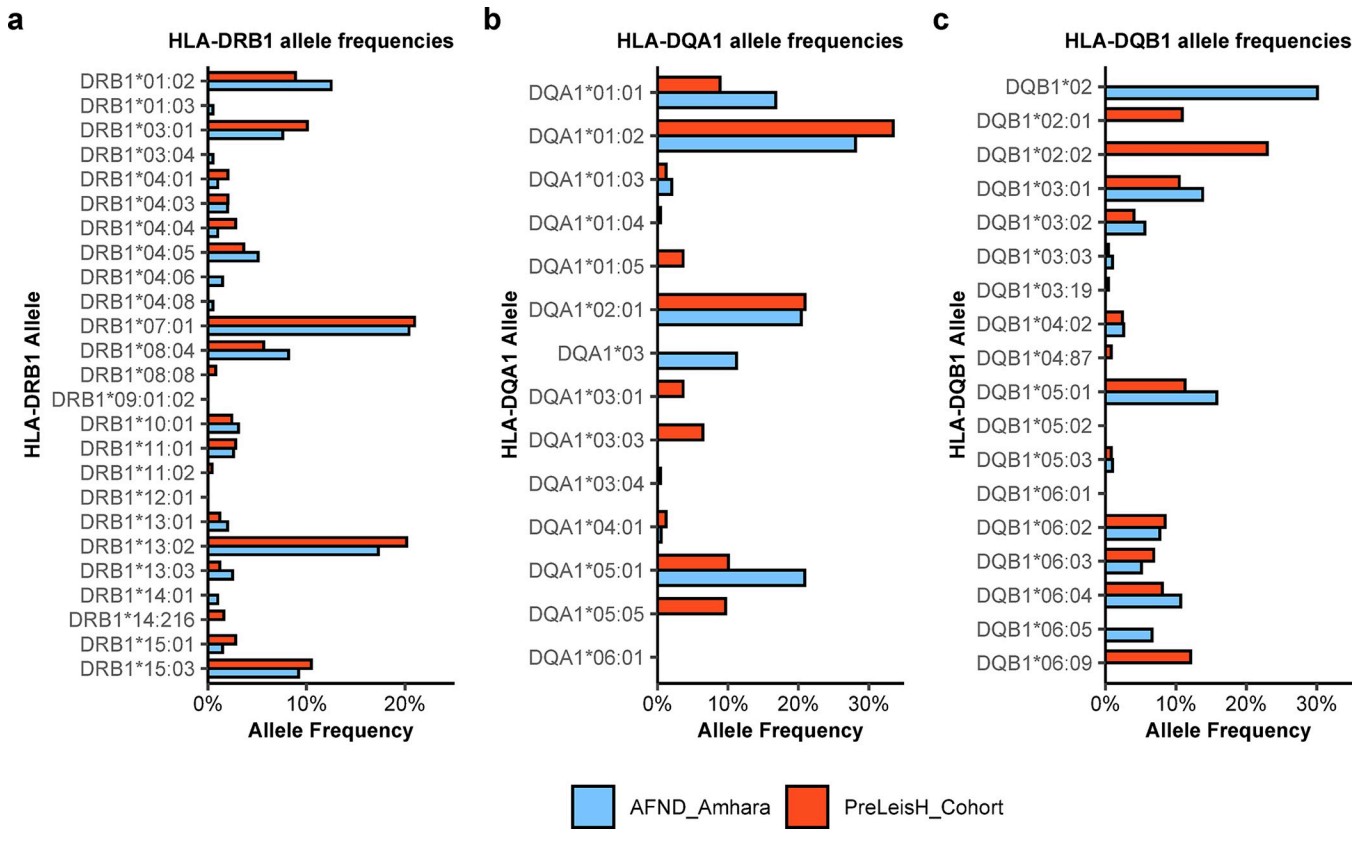

**Fig 3. HLA allele frequencies in the study participants compared to the AFND_Amhara reference for A) HLA-DRB1, B) HLA-DQA1 and C) HLA-DQB1.** The study participants (N = 124) were all adult HIV patients that were either positive for at least two Leishmania infection markers at any timepoint during the study (see methods), or that had a history of visceral leishmaniasis. For HLA-DRB1, HLA-DQA1, and HLA-DQB1, the HLA allele frequencies were compared to those of a prior study in a similar population (also living in the Leishmania-endemic Amhara region of North-West Ethiopia; AFND_Amhara) [25]. The HLA allele frequencies reported by the AFND_Amhara study were determined using sequence-specific oligonucleotides probes, targeting only specific HLA-DRB1, HLA-DQA1, and HLA-DQB1 alleles, while we used the Oxford Nanopore Technologies sequencing-based NanoTYPE assay to detect all alleles spanning 11 different HLA genes.

**Table 4. Description of HLA alleles or allele groups known to be associated with VL development on other continents.** Alleles are described as protective against VL development (OR < 1) or as increasing the risk for VL development (OR > 1). This table was taken and adapted from de Vrij et al., 2021 [15].

| Reference | Protective alleles | Risk alleles | Sample size | Species | Study Location |
|---|---|---|---|---|---|
| Singh et al., 2018 [17] | DRB1*01:01DRB1*15:01DRB1*15:02DRB1*16:02 | DRB1*11DRB1*14:04DRB1*13:01DRB1*13:02 | *Patients N = 889**Control N = 977* | *L. donovani**L. infantum* | IndiaBrazil |
| Fakiola et al., 2013 [16] | DRB1*01DRB1*15DRB1*16 | DRB1*11DRB1*13DRB1*14 | *Three cohorts.**Total Patients N = 2287**Total Control N = 3692* | *L. donovani**L. infantum* | IndiaBrazil |
| Faghiri et al., 1995 [18] |  | A*26 | *Patients N = 52**Control N = 222* | *L. donovani* | Iran |

stratification nor deviation from the Hardy-Weinberg equilibrium. For HLA-DRB1 (**Fig 3A**), -DQA1 (**Fig 3B**), and -DQB1 (**Fig 3C**), we observed allele frequencies generally in line with those observed in the Amhara population by Fort et al. in 1998 (**S2 Table**) [25]. A notable exception, we observed less than half (10%) the allele frequency of HLA-DQA1*05:01 in our study population than those observed by Fort et al. (20.9%).

Next, to verify whether any of the VL development-associated HLA alleles previously identified, in other studies performed in other continents, were not detected in our analysis due to insufficient coverage in the population, we looked at the prevalence of these HLA alleles in our study cohort. HLA alleles already known to be associated with VL development from prior literature are listed in **Table 4**.

As shown in **Table 4**, with the exception of HLA-A*26, all previously described HLA alleles associated with VL development were HLA-DRB1 alleles. The HLA-A*26 allele group has been linked to VL development in Iran, while the HLA-DRB1 alleles have been extensively linked to VL development in Brazil and the Indian subcontinent. In contrast to the studies described in **Table 4**, we did not observe HLA associations for any of these alleles (**S1 Table**). Moreover, all HLA-DRB1 alleles reported in **Table 4** were undetected or below an allele frequency of 0.05 in both our study cohort and the AFND_Amhara cohort, with the exception of HLA-DRB1*13:02 that had an allele frequency of 0.173 in our study population (**Fig 3** and **S2 Table**). While HLA-DRB1*13:02 was sufficiently present in our study population with an allele frequency of 0.173, we could not replicate the finding of Singh et al., 2018, in our Ethiopian adult HIV study population [17]. For HLA-A*26, we detected HLA-A*26:01, which was only present in one participant and not sufficiently present for HLA association analyses.

Taken together, our study population constituted a representative cohort of the Amhara region with a divergent HLA diversity compared to studies on other continents, indicating the need for specific HLA association studies with VL development in Africa.

## Discussion

Identifying risk factors for VL development in PLWH could be imperative for patient stratification in a screen-and-treat approach to tackle VL before disease onset. In Ethiopia, PLWH are placed in an active HIV treatment program with follow-up care and ART monitoring at three to six months intervals, facilitating such a screen-and-treat strategy. In this work, we investigated whether specific HLA alleles were associated with VL development in a NW-Ethiopian *Leishmania*-infected cohort, and identified an association between HLA-A*03:01 and an increased risk for VL development. To our knowledge, this is the first study to address the association of HLA alleles with VL development in Africa, and the first

study globally to investigate this in PLWH. Moreover, we provided HLA allele frequency data that significantly contributes to improved HLA population frequency data in Ethiopia, which until this study, was only publicly available for a select number of HLA class II alleles derived from a single study in 1998 (n = 98) [25]. Finally, we showed that the ONT-sequencing-based NanoTYPE kit has sufficient accuracy for HLA association analyses and could expedite HLA typing in low-resource settings.

Here, we report on a newly described association between HLA*03:01 and an increased risk for VL development in PLWH in North-West Ethiopia. In contrast to our study, earlier work performed in the Fars province of Iran observed no association between HLA-A*03 and VL development, reporting similar frequencies of HLA-A*03 between the VL cases and healthy controls in their study [18]. This can be explained by a number of reasons. Firstly, their HLA genotyping resolution was low (detecting only up to the first field), and consequently, it is uncertain whether HLA*03:01 was sufficiently prevalent, or whether other alleles of the HLA*03 allele group such as HLA*03:02 were more prevalent in their study population. This argument is strengthened by the observation that our present study did not replicate the association between specific alleles of the HLA-DRB1 locus and VL development, which was extensively characterised in India and Brazil, due to insufficient prevalence of these HLA-DRB1 alleles in the study population [16,17]. Secondly, we compared between VL cases and stringently defined asymptomatic *Leishmania*-infected participants, instead of comparing between VL cases and healthy individuals, strengthening our statistical power. The study in Iran that did not observe an association between HLA*03 and VL development, compared seemingly healthy participants with any clinically suspicious VL cases (without microscopic confirmation or infection markers). Healthy individuals do not represent a VL resistant group, and may develop VL once exposed to the parasite. In contrast, the stringently defined and longitudinally monitored asymptomatic *Leishmania*-infected participants in our study are more likely to be resistant to VL development, and thus represent a better control group. While it is still a possibility that some individuals of the asymptomatic *Leishmania*-infected group developed VL after our long-term follow-up, we expect this to be a minor fraction as 40% of asymptomatic *Leishmania* infection marker-positive individuals generally revert to negative markers within 12 months, and in general only a small proportion of asymptomatic individuals actually develop VL [6].

The mechanisms that underpin the association between HLA-A*03:01 and VL development could not be identified within this study. However, several yet-to-be-identified mechanisms may underlie the increased risk of VL development. For instance, HLA-A*03:01 may be less efficient in binding *Leishmania*-derived antigens, subsequently evading host immunity due to impaired antigen presentation to cognate T cells. However, we observed cases of heterozygosity in our study, and in such cases would expect the other allele to compensate for this lack of binding capacity. Another more likely mechanism is that HLA-A*03:01 very efficiently binds *Leishmania* antigens and elicits immunodominance, but rather leads to excessive activation of cytotoxic T cells or inflammatory T cells. In support of this, Singh et al. demonstrated that *in vitro* peptide stimulation of whole blood from cured immunocompetent VL patients homozygous for the risk-associated HLA-DRB1*13:01 allele resulted in a lower IFN-γ to IL10 ratio than in patients homozygous for the protection-associated HLA-DRB1*15:01 allele [17]. In a similar line to the previous mechanism, *Leishmania*, as complex eukaryotic parasite residing and replicating within phagolysosomes of antigen-presenting cells, may have co-evolved with HLA-A*03:01 to evade host immune recognition by producing immunodominant antigens that do not elicit meaningful protective immunity, 'trapping' the host into an incorrect immune response. While all of these factors could explain the association, further mechanistic studies are warranted to identify the correct one. Whether the HIV co-infection impacts the

identified association between HLA-A*03:01 and VL development, could not be answered in this study, and to date, no link between HLA-A*03:01 and HIV infection outcome has been observed.

In addition to the observed association between HLA-A*03:01 and VL development, we identified four alleles (HLA-A*02:01, HLA-C*16:04, HLA-DQA1*01:05, HLA-DQB1*05:01) that were significantly different before multiple testing correction. Of these, HLA-C*16:04 and HLA-DQA1*01:05 were present exclusively in participants with VL history. While no specific HLA-DQA1 or HLA-DQB1 allele has been described to be associated with VL development, single nucleotide polymorphisms (SNPs) in the HLA-DQA1 region have been linked to VL development susceptibility in India and Brazil [16]. Furthermore, HLA-DQB1*05:01 positivity was well correlated with the SNPs that increased VL development susceptibility. Despite the lack of significance after multiple testing correction, the 95% CI of HLA-A*02:01 spans 0.18–0.98 and does not exceed 1. While this could indicate a weaker association of HLA-A*02:01 with protection against VL development that was masked by overcorrection, it may also be an artefact due to the enrichment of HLA-A*03:01 in the past VL group. It is of note that reducing the analysis resolution to the 2-digit level, to increase the statistical power, provided identical results to the current 4-digit level analyses. Thus, whether these are truly associated with VL development and could have prognostic value, or are just an artefact of the relative group sizes, would need validation in a larger cohort. None of the observed significantly different HLA alleles in this study on VL matched the previously described cutaneous leishmaniasis-associated HLA alleles [15]. However, these studies were mostly conducted in South-America which may not share HLA allele occurrence, with the exception of a study in Egypt, and were all conducted on CL-causing species, which may not share common antigens with VL-causing species.

While HLA-A*03:01, with a prevalence of 43.6% in participants with VL history, may have little value as a sole predictor of VL development, it can have substantial prognostic value in a larger clinical prediction algorithm, together with, for example, patterns of *Leishmania* infection markers, to accurately predict VL onset risk. Moreover, HLA-A*03:01 as an invariant genetic risk factor would only need to be measured once, and can thus contribute greatly to initial patient stratification into higher and lower risk groups in a screen-and-treat approach. Yet, it remains to be explored whether a combination of different HLA alleles, including HLA*03:01, *Leishmania* infection markers, or other host or HIV-related risk factors such as CD4 counts, could sufficiently predict VL onset or contribute to patient stratification into risk groups for monitoring or a screen-and-treat approach. In this study, we could not identify any haplotype association with VL development as most haplotypes were unique to individuals, and it would require a larger cohort study to identify whether a combination of HLA alleles predispose to VL disease. As an ancilliary study on biobanked samples, we could not include any incident VL cases, from which samples were used in the primary analyses. Including the incident VL cases would have allowed us to explore whether the combination of HLA genotype, the pattern of *Leishmania* markers, and other risk factors such as CD4 counts, could accurately predict VL onset. Such prospective studies are also needed to adjust for CD4 counts and HIV viral loads at the time of VL development to exclude confounding of the observed association. Further validation of the association between HLA alleles and VL development in NW-Ethiopia is warranted, and should include mechanistic investigations to study how variation in the HLA region could lead to a higher risk of VL development, in new prospective studies (such as the planned Clinical Prognostic Score to Predict Relapse in VL (CPS) study https://clinicaltrials.gov/study/NCT05602610), which includes both VL and VL-HIV patients. Including both VL-HIV patients and VL patients without HIV co-infection may provide more insight in whether the HIV co-infection affects the identified HLA association. Another

**Table 5. Advantages and disadvantages of a conventional Illumina sequencing-based HLA genotyping assay versus an ONT sequencing-based HLA genotyping assay.**

| HLA genotyping method | Advantages | Disadvantages |
|---|---|---|
| AlloSeq Tx17 (CareDx; Illumina-based) | • Includes all exons for all genes, improving HLA allele calling and resolving ambiguities.<br>• Up to the 4th field (8-digit) resolution.<br>• Also includes the HLA-E, -F, -G, -H, MICA, and MICB genes. | • Time-intensive (days instead of hours)<br>• No real-time sequencing<br>• Need for high capital investment (Illumina sequencing devices)<br>• Short-read data |
| NanoTYPE (Omixon; ONT-based) | • Turnaround time of hours instead of days<br>• User-friendly workflow with minimal time spent preparing library<br>• Low capital investment (can run on an ONT MinION device of ~1000USD)<br>• Portable (MinION Mk1B device is palm-sized)<br>• Sequencing can be monitored and stopped in real-time once sufficient coverage has been reached | • Limited to the 3rd field (6-digit) resolution<br>• Does not include all exons (more ambiguities)<br>• Does not include all HLA region genes |

limitation is that we could not differentiate between *Leishmania* species in the main study, although the main etiological agent of VL in Ethiopia is *L. donovani*. Different species have various genomic alterations that could skew the antigen-binding repertoire, and thus have different HLA associations. A final limitation is that we did not assess prior exposure of participants to cutaneous leishmaniasis (CL)-causing Leishmania species nor CL scars, but due to vector dynamics in Ethiopia, we do not expect the participants to have been widely exposed to CL.

To expedite HLA genotyping in resource-constrained countries like Ethiopia, with a hard-to-reach population, we require easy-to-use and quick methods with low initial capital investment. In the past, this has proven challenging as HLA typing was either laborious (in case of PCR with sequence-specific oligonucleotide probes) or required high capital investments (such as Illumina-based sequencing). Here, we showed that ONT-based HLA genotyping holds several advantages such as a fast turnaround time (hours instead of days) and low capital investment, making it suitable for application in a low-resource setting (**Table 5**). We believe the described ONT-based HLA typing assay will make grand-scale HLA typing in low-and-middle income countries more feasible, and can contribute to replication and validation of the association between specific HLA alleles and disease in these resource-constrained settings.

## Conclusions

We demonstrated an association between the HLA-A*03:01 allele and an increased risk for VL development in PLWH in NW-Ethiopia. As such, it holds promise as a potential predictor of VL onset in HIV patients with *Leishmania* infection. However, larger cohort studies studying a clinical prediction algorithm are required to replicate and expand upon these findings, adjust for confounders, and to find out the mechanisms that underpin the identified associations. Finally, we argue that ONT-based HLA genotyping may significantly expedite such validation studies in resource-constrained settings, to enable future inclusion of HLA alleles in clinical stratification algorithms to predict VL onset in the highly burdened Ethiopian setting and beyond.

## Supporting information

**S1 Fig. HLA allele frequencies in the study participants for A) HLA-A, B) HLA-B, C) HLA-C, D) HLA-DPA1, E) HLA-DPB1, and F) HLA-DRB3/4/5.** The study participants

(N = 124) were all adult HIV patients that were either positive for at least two Leishmania infection markers at any timepoint during the study (see methods), or that had a history of visceral leishmaniasis. We used the Oxford Nanopore Technologies sequencing-based Nano-TYPE assay to detect all alleles spanning 11 different HLA genes. HLA-DRB1, HLA-DQA1, and HLA-DQB1 are listed in Fig 3 of the main manuscript.
(TIF)

**S2 Fig. HLA class I allele frequencies in the study participants per participant group.** The study participants (N = 124) were all adult HIV patients that were either positive for at least two *Leishmania* infection markers at any timepoint during the study (No VL History, see methods), or that had a history of visceral leishmaniasis (VL history). We used the Oxford Nanopore Technologies sequencing-based NanoTYPE assay to detect all alleles spanning 11 different HLA genes.
(TIF)

**S3 Fig. HLA class II allele frequencies in the study participants per participant group.** The study participants (N = 124) were all adult HIV patients that were either positive for at least two *Leishmania* infection markers at any timepoint during the study (No VL History, see methods), or that had a history of visceral leishmaniasis (VL history). We used the Oxford Nanopore Technologies sequencing-based NanoTYPE assay to detect all alleles spanning 11 different HLA genes.
(TIF)

**S1 Table. All HLA alleles included in the HLA association analysis between past VL developers (n = 78) and asymptomatic Leishmania controllers (n = 46), ranked ascendingly by lowest p-value after Benjamini-Hochberg multiple testing correction.** An OR above 1 indicates an increased risk for VL development, while an OR below 1 indicates increased protection against VL development.
(DOCX)

**S2 Table. The HLA allele frequencies of all HLA alleles detected in our study population of 124 Leishmania-infected and HIV co-infected individuals living in NW-Ethiopia.** All alleles are reported up to the second field resolution (4-digit). Counts are the number of times an allele appears in the population.
(DOCX)

## Acknowledgments

We want to thank Tessa de Block and the Clinical Virology Unit at ITM for their generous support in ONT flow cells. We thank the Clinical Reference Laboratory of the Institute of Tropical Medicine for providing access to their facilities and equipment, and the ITM biobank staff for their support. Finally, we would like to thank all the study participants and the staff involved in the study at the Abdurafi Health Center and the Gondar Leishmaniasis Research and Treatment Center.

## Author Contributions

**Conceptualization:** Nicky de Vrij, Wim L. Cuypers, Thao-Thy Pham, Wim Adriaensen.

**Data curation:** Nicky de Vrij, Tadfe Bogale.

**Formal analysis:** Nicky de Vrij, Romi Vandoren.

**Funding acquisition:** Nicky de Vrij, Koert Ritmeijer, Kris Laukens, Bart Cuypers, Ermias Diro, Johan van Griensven, Wim Adriaensen.

**Investigation:** Nicky de Vrij, Kadrie Ramadan, Anke Van Hul, Ann Ceulemans, Arega Yeshanew, Hailemariam Beyene, Kasaye Sisay, Aderajew Kibret, Dagnew Mersha.

**Methodology:** Nicky de Vrij, Thao-Thy Pham, Pieter Meysman, Wim Adriaensen.

**Project administration:** Nicky de Vrij, Mekibib Kassa, Roma Melkamu, Tadfe Bogale, Rezika Mohammed.

**Resources:** Nicky de Vrij, Anke Van Hul, Mekibib Kassa, Roma Melkamu, Bart Cuypers, Wim Adriaensen.

**Software:** Nicky de Vrij.

**Supervision:** Pieter Meysman, Kris Laukens, Bart Cuypers, Wim Adriaensen.

**Visualization:** Nicky de Vrij.

**Writing – original draft:** Nicky de Vrij, Romi Vandoren.

**Writing – review & editing:** Nicky de Vrij, Wim L. Cuypers, Florian Vogt, Saskia van Henten, Thao-Thy Pham, Pieter Meysman, Bart Cuypers, Ermias Diro, Johan van Griensven, Wim Adriaensen.

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
