## [Decision Letter · Decision Letter 0]

11 Jun 2024

Dear Dr. Adriaensen,

Thank you very much for submitting your manuscript "HLA-A*03:01 is associated with visceral leishmaniasis development in people living with HIV in Ethiopia" for consideration at PLOS Neglected Tropical Diseases. As with all papers reviewed by the journal, your manuscript was reviewed by members of the editorial board and by several independent reviewers. In light of the reviews (below this email), we would like to invite the resubmission of a significantly-revised version that takes into account the reviewers' comments. 

We cannot make any decision about publication until we have seen the revised manuscript and your response to the reviewers' comments. Your revised manuscript is also likely to be sent to reviewers for further evaluation.

Sincerely,

Kelly Hawley

Academic Editor

Daniel Masiga

Section Editor

Reviewer's Responses to Questions

**Key Review Criteria Required for Acceptance?**

**Methods**

-Are the objectives of the study clearly articulated with a clear testable hypothesis stated?

-Is the study design appropriate to address the stated objectives?

-Is the population clearly described and appropriate for the hypothesis being tested?

-Is the sample size sufficient to ensure adequate power to address the hypothesis being tested?

-Were correct statistical analysis used to support conclusions?

-Are there concerns about ethical or regulatory requirements being met?

Reviewer #1: 1- Does the biobank consent exclusively cover consent for genetic testing? If feasible, please provide a copy of the consent form in the supplementary data for transparency and reference.

2- Please ensure that all software used for data analysis is properly referenced.

3- Were the authors' study participants limited to a single ethnic group in Ethiopia, or were patients recruited from various ethnic backgrounds such as Oromos, Amharas, Somalis, etc.? Please provide further information in the manuscript.

4- It's crucial to perform a Hardy-Weinberg test on control samples to assess potential population stratification.

5- Did the authors conduct an HLA haplotype association test?

6- Considering weaker associated HLA alleles, have you considered using 2-digit alleles to increase the sample size in each comparison?

Reviewer #2: 4. The selection of study groups may introduce noise to the real association of HLA variants and disease progression. Although the team is contrasting patients who had experienced VL, vs. those that are asymptomatic, it is unknown whether the VL group was asymptomatic prior to developing disease. No prospective follow-up of this cohort was done prior to onset of symptoms, which begs the question of their infection status prior to developing VL. To really identify markers associated with disease progression, the prospective cohort should start from all asymptomatic patients, with rigorous follow-up to identify onset of symptoms in a given time period. That would clearly define the study groups. 

5. It is unclear how the sample size is suitable for the proposed analyses. A total of 83 alleles were analyzed in a sample size of 124 patients. The likelihood for spurious associations is very high. The authors mention in the methods that a previous study with some HLA frequencies in Ethiopian populations. Why not use this at least as a starting point for sample size estimation? Please include proper sample size estimations and the statistical support for the effect given the sample size and multiple testing. 

6. Are there effects associated with possible admixture of the populations? No description of the ethnic background was included for this cohort. Please include in the demographics table description of ethnicity/race.

7. There is over-interpretation od data, for example in the results section “Taken together, the strong association between HLA-A*03:01 and VL development, and the high prevalence of HLA-A*03:01 in the past VL group, suggests prognostic value in capturing those high at risk for VL in a clinical algorithm”. How can this be ascertained if no proper sample size estimations, control for confounding variables such as genetic background, among others, were made?

- Protocol approval numbers should be included

- Datasets could be made available in a de-identified manner, including HLA genotypes and at least group calssifications.

**Results**

-Does the analysis presented match the analysis plan?

-Are the results clearly and completely presented?

-Are the figures (Tables, Images) of sufficient quality for clarity?

Reviewer #1: (No Response)

Reviewer #2: 1. It is not clear what the authors consider a VL case; VL patients with clinical manifestations, or symptomatic plus asymptomatic cases. This should be clearly stated because the epidemiological data they present (i.e. first paragraph of the introduction) can be misinterpreted. 

2. In an area where CL is also quite prevalent, how does “VL asymptomatic infections” are differentiated from pre-exposure to CL causing species in those that are rK39 negative?

3. In the intro the authors comment: “The asymptomatic stage preceding VL disease is detectable by a variety of Leishmania infection markers, and can thus provide an opportune moment to screen for those at risk for VL development and to initiate preventative strategies (4, 7, 8)”. Please define which are these methods and their relevance in discriminating active infection, immunological memory of infection, and pre-exposure to different Leishmania species.

**Conclusions**

-Are the conclusions supported by the data presented?

-Are the limitations of analysis clearly described?

-Do the authors discuss how these data can be helpful to advance our understanding of the topic under study?

-Is public health relevance addressed?

Reviewer #1: (No Response)

Reviewer #2: 8. Is there any association between the HLA-A*03:01 and the outcome of HIV infection? How does having HIV affect the development of VL according to HLA? Is this same difference expected to occur in a non-HIV populations. All these aspects need to be discussed in the paper.

**Editorial and Data Presentation Modifications?**

Reviewer #1: 1- In Figure 3, please provide separate bar-plots for allele frequencies of past VL and the asymptomatic cohort.

Reviewer #2: (No Response)

**Summary and General Comments**

Reviewer #1: Overall, the manuscript is well-written. Particularly intriguing is the comparison between ONT-based and short-read-based HLA typing, as the application of ONT-based HLA typing could be pivotal in Low- and Middle-Income Countries (LMICs).

Reviewer #2: I strongly suggest to change the title, and hone down the asseverations of associations being made. This study provides some preliminary findings that may guide larger cohort studies that are properly powered to address the relevant question being asked.

PLOS authors have the option to publish the peer review history of their article (what does this mean?). If published, this will include your full peer review and any attached files.

Reviewer #1: No

Reviewer #2: Yes: Maria Adelaida Gómez
---

## [Decision Letter · Decision Letter 1]

23 Aug 2024

Dear Dr. Adriaensen,

Thank you very much for submitting your manuscript "A preliminary indication that HLA-A*03:01 may be associated with visceral leishmaniasis development in people living with HIV in Ethiopia" for consideration at PLOS Neglected Tropical Diseases. As with all papers reviewed by the journal, your manuscript was reviewed by members of the editorial board and by several independent reviewers. The reviewers appreciated the attention to an important topic. Based on the reviews, we are likely to accept this manuscript for publication, providing that you modify the manuscript according to the review recommendations. 

Sincerely,

Kelly Hawley

Academic Editor

Claudia Brodskyn

Section Editor

Reviewer's Responses to Questions

**Key Review Criteria Required for Acceptance?**

**Methods**

-Are the objectives of the study clearly articulated with a clear testable hypothesis stated?

-Is the study design appropriate to address the stated objectives?

-Is the population clearly described and appropriate for the hypothesis being tested?

-Is the sample size sufficient to ensure adequate power to address the hypothesis being tested?

-Were correct statistical analysis used to support conclusions?

-Are there concerns about ethical or regulatory requirements being met?

Reviewer #1: (No Response)

Reviewer #3: Overall, the manuscript is well-qualified and has novelty. However, several flaws must adequately be taken care of.

In the M&M, Fig 1 should be explained in detail.

Fig 1 should be edited graphically and have harmony in their color classification. For example, choose a specific color for No sample available, and so on. 

Don’t use italics in the flowchart. They should edit the fellow chart entirely based on the same literature in the PLOS NTD. 

The inclusion criteria and exclusion criteria should be defined in the text.

How did they recognize asymptomatic cases? 

The asymptomatic N: 126 was not included; explain it in the manuscript.

Please mention the statistical analysis in detail. How did they evaluate the normality of the quantitative variables in each group?

**Results**

-Does the analysis presented match the analysis plan?

-Are the results clearly and completely presented?

-Are the figures (Tables, Images) of sufficient quality for clarity?

Reviewer #1: (No Response)

Reviewer #3: As they used PCR, they should mention the etiological agent of VL. I can’t find the species of VL in the results.

 Did they evaluate viscerotropic CL that can be detected in HIV-positive patients?

**Conclusions**

-Are the conclusions supported by the data presented?

-Are the limitations of analysis clearly described?

-Do the authors discuss how these data can be helpful to advance our understanding of the topic under study?

-Is public health relevance addressed?

Reviewer #1: (No Response)

Reviewer #3: Discussion:

Please mention the Leishmaniasis-associated HLA alleles derived from the literature in the discussion. Were there any differences between VL caused by L. infantum or L. donovani and also CL, MCL, DCL, or VL caused by L. tropica? Did the author find the alleles in common, especially in viscerotropic infections or severity of infection, symptomatic and asymptotic forms? Please explain them in the discussion.

**Editorial and Data Presentation Modifications?**

Reviewer #1: (No Response)

Reviewer #3: (No Response)

**Summary and General Comments**

Reviewer #1: The authors have addressed all my comments satisfactorily. I recommend this manuscript for publication in PLOS Neglected Tropical Diseases.

Reviewer #3: (No Response)

PLOS authors have the option to publish the peer review history of their article (what does this mean?). If published, this will include your full peer review and any attached files.

Reviewer #1: No

Reviewer #3: No

Figure Files:

Data Requirements:

Reproducibility:

References

---

## [Editor Report · Decision Letter 2]

18 Sep 2024

Dear Dr. Adriaensen,

We are pleased to inform you that your manuscript 'A preliminary indication that HLA-A*03:01 may be associated with visceral leishmaniasis development in people living with HIV in Ethiopia' has been provisionally accepted for publication in PLOS Neglected Tropical Diseases.

Best regards,

Kelly Hawley

Academic Editor

Claudia Brodskyn

Section Editor

---

## [Editor Report · Acceptance letter]

23 Sep 2024

Dear Dr. Adriaensen,

We are delighted to inform you that your manuscript, "A preliminary indication that HLA-A*03:01 may be associated with visceral leishmaniasis development in people living with HIV in Ethiopia," has been formally accepted for publication in PLOS Neglected Tropical Diseases.

Best regards,

Shaden Kamhawi

co-Editor-in-Chief

Paul Brindley

co-Editor-in-Chief
